# Transcriptomic Analyses Reveal the Effects of Walnut Kernel Cake on Adipose Deposition in Pigs

**DOI:** 10.3390/genes15060667

**Published:** 2024-05-23

**Authors:** Lei Liu, Xiaodan Shang, Li Ma, Dawei Yan, Adeyinka Abiola Adetula, Ying Bai, Xinxing Dong

**Affiliations:** 1College of Animal Science and Technology, Yunnan Agricultural University, Kunming 650201, China; liulei03@caas.cn (L.L.); 1995027@ynau.edu.cn (D.Y.); 2Shenzhen Branch, Guangdong Laboratory of Lingnan Modern Agriculture, Key Laboratory of Livestock and Poultry Multi-omics of MARA, Agricultural Genomics Institute at Shenzhen, Chinese Academy of Agricultural Sciences, Shenzhen 518000, China; 3School of Life Sciences and Food Engineering, Hebei University of Engineering, Handan 056038, China; 17349805197@163.com; 4Department of Animal Husbandry and Veterinary Medicine, Yunnan Vocational and Technical College of Agriculture, Kunming 650212, China; yandaweiynau@163.com; 5Reproductive Biotechnology, Department of Molecular Life Sciences, TUM School of Life Sciences, Technical University Munich, 85354 Freising, Germany; quueenyk10@yahoo.com

**Keywords:** walnut kernel cake, transcriptome, single-cell transcriptome, adipose deposition, pig

## Abstract

With the rising cost of animal feed protein, finding affordable and effective substitutes is crucial. Walnut kernel cake, a polyphenol-, fiber-, protein- and fat-rich byproduct of walnut oil extraction, has been underexplored as a potential protein replacement in pig feed. In this study, we found that feeding large Diqing Tibetan pigs walnut kernel cake promoted adipose deposition and improved pork quality during pig growth. Transcriptome analysis revealed the upregulation of genes *ANGPTL8*, *CCNP*, *ETV4*, and *TRIB3*, associated with adipose deposition. Pathway analysis highlighted enrichment in adipose deposition-related pathways, including PPAR, insulin, PI3K-Akt, Wnt, and MAPK signaling. Further analysis identified DEGs (differentially expressed genes) positively correlated with adipose-related traits, such as *PER2* and *PTGES*. Single-cell transcriptome data pointed to the specific expression of *CD248* and *PTGES* in adipocyte progenitor/stem cells (APSCs), pivotal for adipocyte differentiation and adipose deposition regulation. This study demonstrates walnut kernel cake’s potential to substitute soybean cake in pig feed, providing high-quality protein and promoting adipose deposition. It offers insights into feed protein replacement, human functional food, fat metabolism, and related diseases, with marker genes and pathways supporting pig breeding and pork quality improvement.

## 1. Introduction

As nutritious food, research on walnuts in the human diet has always attracted much attention [1,2]. Multiple studies have linked walnut intake to cardiovascular health. Walnuts are rich in unsaturated fatty acids, phytosterols and fiber, which can help lower cholesterol levels, improve blood lipid metabolism, and reduce the risk of cardiovascular disease [3,4,5]. Walnuts are rich in antioxidants, such as vitamin E, polyphenols and antioxidant enzymes, which help neutralize free radicals, reduce oxidative stress damage, and improve the body’s antioxidant capacity [6]. Some studies suggest that walnut consumption may help improve cognitive function and brain health [7]. Components such as omega-3 fatty acids and antioxidants in walnuts are thought to be beneficial to brain function [8,9]. However, perhaps due to the small global cultivation range and production issues, walnut and its ancillary products have been less studied in the field of livestock farming. China’s walnut planting area and output rank first in the world, and it is also the country with the highest walnut consumption in the world [10]. Yunnan Province is the largest walnut-producing area in China (accounting for 27.17% of the national walnut production), and Fengqing County is the main walnut-producing area in Yunnan Province [11]. A large amount of walnut cake is produced here every year due to oil extraction. In line with a sustainable path, the concept of ‘waste to wealth’ leading to ‘green growth’ is a great opportunity to improve food security and is being adopted by many developed and developing countries [12].

In the pig breeding industry, seeking efficient and economical feed alternatives and improving meat quality have always been the focus of research. Adjustment of feed ingredients can not only reduce production costs but also improve pig growth performance and meat production quality. Walnut cake is the solid residue left over from the oil extraction process from walnuts and is often used as feed or for other agricultural purposes. It has attracted attention as a potential feed substitute because it is rich in protein and fat and has potential nutritional advantages [2]. Research on walnut kernel cake may provide new feed options for the pig industry and improve the health and nutrition of animal husbandry [13]. Partial replacement of soybean meal with walnut cake as a protein source has an impact on the meat quality of broiler breast meat [14,15]. However, there are relatively few studies on the application of walnut cake in pig feed and its effect on pig fat deposition, especially on the Diqing Tibetan pig, a local breed in Yunnan Province, China [16]. Therefore, exploring the effect of walnut cake on pig fat deposition has important theoretical and application value.

This study aimed to investigate the effect of walnut kernel cake as a dietary protein substitute on adipose deposition-related traits in Diqing Tibetan pigs and to explore its molecular mechanism at the transcriptome level. We initially investigated the effects of walnut kernel cake on adipose-related traits at different growth and developmental stages of pigs. Subsequently, we dissected the molecular mechanisms underlying the alterations of adipose-related traits induced by walnut kernel cake at the transcriptome level. Finally, we further elucidated changes in adipocyte types and their marker genes at the single-cell transcriptome level during this process. This study provides new insights and references for research on feed protein substitution, pig breeding and genetic improvement of meat quality.

## 2. Materials and Methods

### 2.1. Experimental Treatment and Trait Recording

Twelve large Diqing Tibetan pigs at 90 days of age (initial body weight = 8.90 ± 1.85 kg) were used in this experiment conducted in Yunnan (Yunnan Province, China). The 12 pigs were allotted to two dietary treatments and two stages (three pigs/diet/stage). Diets included a corn–soybean cake basal diet (Feed A) and a walnut kernel cake diet (Feed B) containing 5% expeller-pressed walnut kernel cake substituted for corn and soybean cake (Figure 1). Pigs had ad libitum access to feed and water at all times. The nutritional value of walnut kernel cake is shown in Table 1, which shows that walnut kernel cake is a valuable source of crude protein. The nutritional value of soybean meal is shown in Appendix A. The ingredients and nutritional value of the diets are shown in Table 2.

The slaughter weight of fattening pigs is around 100 kg. We collected adipose tissue from fattening pigs before and after they were slaughtered to study the effects of different feeds on pig adipose deposition. When the body weight reached 80 kg, the backfat tissues of three pigs fed with Feed A and the backfat tissues of three pigs fed with Feed B were collected (three pigs/diet/stage). This was also performed when the pigs reached 120 kg (three pigs/diet/stage). Samples were taken immediately after euthanasia, frozen in liquid nitrogen, and stored at −80 °C until used for RNA extraction and sequencing (Figure 1). Meanwhile, the adipose-related traits were recorded according to the technical regulation for testing of carcass traits in lean-type pig (NY/T 825-2004), containing caul fat rate (CFR), abdominal fat rate (AFR), backfat thickness at three positions (BF_A, BF_B, BF_C), average backfat thickness (BF_Avg), backfat thickness between the 6th and 7th ribs (BF_67), fat weight rate at forequarters, middle torso and hindquarters of pig (FWR_F, FWR_M, FWR_H), and total fat weight rate (FWR_T).

### 2.2. RNA Extraction, Library Construction, and Sequencing

We extracted total RNA from these 12 samples using Trizol reagent (Invitrogen, Waltham, MA, USA) as follows: Transfer 50–100 mg of adipose tissue into a 2 mL centrifuge tube, add 1 mL Trizol reagent and RNAase-free steel beads. Thoroughly mix the sample and reagent using a homogenizer, and let it stand at room temperature for 5 min. Add 0.2 mL chloroform, shake for 15 s, and let it stand for 2 min. Centrifuge at 4 °C, 12,000× *g* 15 min, and collect the supernatant. Add 0.5 mL isopropanol, gently mix the liquid in the tube, and let it stand at room temperature for 10 min. Centrifuge at 4 °C, 12,000× *g* 10 min, and discard the supernatant. Add 1 mL of 75% ethanol to wash the precipitate gently. Centrifuge at 4 °C, 7500× *g* 5 min, and discard the supernatant. Air dry the pellet, and dissolve it in an appropriate amount of DEPC-treated water. Then treat the RNA with DNase I enzyme (Thermo Fisher Scientific, Waltham, MA, USA) to remove DNA contamination. The RNA information of the 12 samples is shown in Appendix A.

The purity, concentration and integrity of the RNA were tested by NanoDrop spectrometer (Thermo Fisher Scientific), Agilent 2100 Bioanalyzer (Agilent, Santa Clara, CA, USA), and Qubit 2.0 fluorescence analyzer (Thermo Fisher Scientific). TruSeq Stranded Total RNA Sample Prep Kit (Illumina, San Diego, CA, USA) was used for RNA library construction. First, poly(A)+ RNA is enriched using magnetic beads, and then the RNA is fragmented using fragmentation buffer. Next, first-strand cDNA was synthesized using SuperScript III reverse transcriptase, and DNA polymerase I and RNase H were used for second-strand synthesis. Afterwards, we performed end repair, A-tail modification and adapter ligation, and used PCR amplification to amplify the library to the desired concentration. Finally, the 12 libraries were subjected to pair-end 150 bp high-throughput sequencing using the Illumina HiSeq X Ten platform.

### 2.3. Sequencing Data Filtering and Differentially Expressed Gene (DEG) Analysis

We first checked the quality of the sequencing data using FastQC (v0.11.8) and then used fastp (v0.20.0) software to remove low-quality sequences with default parameters to obtain clean data [17,18]. The clean reads were further aligned to the reference genome (Sscrofa11.1.97) using STAR (v2.7.1) to compare with the reference genome [19]. The quality values and the mapping rate of RNA-seq data for the 12 samples are shown in Appendix A. We used featureCounts (v2.0.0) to calculate the gene expression level [20], and filtered out genes with an average count of less than 1 in all samples. We then used DESeq2 (v1.38.3) software to normalize gene expression levels, and further analyze differentially expressed genes (DEGs) [21]. The genes with the criterion of |log_2_ (fold change)| > 1 (|log_2_FC| > 1) and unadjusted *p* < 0.05 were considered DEGs.

### 2.4. Functional Enrichment Analysis

Gene function enrichment analysis is a method used to discover gene functions associated with biological processes or pathways in a set of genes. In this study, we took the list of DEGs as input and used the R package clusterProfiler (v4.6.2) to perform Gene Ontology (GO) and Kyoto Encyclopedia of Genes and Genomes (KEGG) functional enrichment analyses to explore the biological functions of DEGs [22]. For GO analysis, we performed the analysis on three different GO categories (molecular function, biological process, and cellular component), setting *p* < 0.05 as the significance threshold. At the same time, we also analyzed all KEGG pathways and set *p* < 0.05 as the significance threshold. We used the wordcloud (v2.6) package and enrichplot (v1.18.4) package in the R language (v4.2.2) to visualize the results of the GO and KEGG enrichment analyses. Significantly enriched pathways and GO terms were shown using barplot and dotplot. Frequency and classification of pathways and GO terms were shown using wordcloud and treeplot. Heatplot and cnetplot functions were also used to generate heatmaps and network diagrams to more intuitively display the relationship and interaction between DEGs and GO or KEGG pathways.

### 2.5. Gene Set Enrichment Analysis (GSEA)

GSEA (Gene Set Enrichment Analysis) is a method commonly used in transcriptome analysis, which can compare gene expression profiles between different samples and find gene sets related to certain biological processes, diseases, etc. [23]. In this study, GSEA was performed using all genes for the two growth stages, respectively. First, all genes were sorted in descending order according to log_2_FC, and then the gseGO and gseKEGG functions of the R package clusterProfiler were used to perform the GSEA of GO and KEGG, respectively. We screened for significantly enriched gene sets with |NES| > 1, *p* < 0.05, and *p*.adjust < 0.25.

### 2.6. Weighted Gene Co-Expression Network Analysis (WGCNA)

We further performed WGCNA in the R software (v4.0.2) environment using the WGCNA package (v1.72) [24]. We filtered the raw data of all 12 RNA-seq samples in this study, eliminated low-expression genes, retained genes with a coefficient of variation greater than 0.4 and 30% of the samples had an expression level greater than 1 to obtain an expression matrix. After normalizing the expression matrix, we used the pickSoftThreshold function to estimate the soft threshold parameter and selected the best parameter value that could achieve a high degree of modularity. Afterwards, using the blockwiseModules function (minModuleSize = 30, mergeCutHeight = 0.4), we clustered the gene expression data and identified co-expressed modules. We used the similarity based on module eigenvectors (Module Eigengene) to calculate the similarity between modules. Through hierarchical clustering, we aggregated highly related modules together and assigned each module a color name. Finally, we imported adipose-related traits and performed correlation analyses with each module to identify key modules that affect the adipose traits.

### 2.7. Correlation Analysis of Adipose-Related Traits and Genes

In order to mine the genes that significantly affect adipose traits, we first extracted the expression level of all 182 genes in the tan module that were highly correlated with adipose traits. We then performed Pearson correlation analysis between the genes and the 11 adipose-related traits, such as Caul fat rate (CFR), Abdominal fat rate (AFR), Backfat thickness_A (BF_A), Backfat thickness_B (BF_B), Backfat thickness_C (BF_C), Backfat thickness_Avg (BF_Avg), Backfat thickness_67 (BF_67), Fat weight rate_Forequarters (FWR_F), Fat weight rate_Middle torso (FWR_M), Fat weight rate_Hindquarters (FWR_H) and Fat weight rate_Total (FWR_T). It was considered to be significantly correlated between genes and traits when *p* < 0.05. Furthermore, we also used the 80 kg and 120 kg stages of DEG expression levels and adipose traits to conduct correlation analyses to explore the molecular mechanism of walnut kernel cake affecting pig adipose deposition in different periods.

### 2.8. scRNA-seq Analysis of Adipose Tissue

To study the effect of walnut kernel cake on pig adipose deposition from the single-cell level, we downloaded the expression matrix of single-cell data (GSE193975) of pig adipose from the NCBI public database [25]. We first used Seurat (v4.3.0) software to remove low-quality cells and genes (nCount_RNA < 1000 and nFeature_RNA < 500 and percent.mt > 20), then used decontX (v4.3.0) and DoubletFinder (v4.3.0) to remove ambient RNA and doublets, respectively [26,27,28]. Subsequently, Seurat software (v4.3.0.1) was used for data dimensionality reduction and clustering. The FindMarkers function in Seurat was used to find the DEGs among different clusters. The DEGs were determined according to the threshold of log_2_FC > 0.25 and *p*.adjust < 0.05. We annotated the cell types of clusters based on classic marker genes and online databases such as Cellmarker (http://bio-bigdata.hrbmu.edu.cn/CellMarker/, accessed on 7 March 2023) and PanglaoDB (https://panglaodb.se/index.html, accessed on 7 March 2023). Moreover, we used Seurat’s DotPlot function to visualize the expression patterns of cell type-specific marker genes. The expression patterns of key genes significantly associated with adipose traits in different cell types were visualized using the VlnPlot function.

### 2.9. Statistical Analysis

Differences in adipose-related traits (CFR, AFR, BF_A, BF_B, BF_C, BF_Avg, BF_67, FWR_F, FWR_M, FWR_H and FWR_T) across comparison combinations were analyzed using a two-tailed *t*-test. Pearson correlation analysis of these traits and gene expression was performed using the “cor” function in the R package with the method of “Pearson”. These genes include the 182 genes in the tan module of WGCNA, DEGs of Feed A and Feed B groups during the 80 kg body weight period, and DEGs of Feed A and Feed B groups during the 120 kg body weight period. The correlation coefficient and *p* value were calculated with default parameters. A *p* value less than 0.05 was considered statistically significant.

## 3. Results

### 3.1. Effects of Walnut Kernel Cake on Traits Related to Adipose Deposition

In order to study the effect of walnut kernel cake on adipose deposition in pigs, we measured 11 traits, such as CFR, AFR, BF_A, BF_B, BF_C, BF_Avg, BF_67, FWR_F, FWR_M, FWR_H and FWR_T at 80 kg and 120 kg, respectively (Figure 2). It was found that at the 80 kg stage, walnut kernel cake significantly increased the CFR of pigs (*p* < 0.05, Figure 2A). At the 120 kg stage, walnut kernel cake significantly increased the BF_67 and FWR_T of pigs (*p* < 0.05, Figure 2B). There was also an increasing trend for traits such as AFR, BF_A, BF_B, BF_C, BF_Avg, FWR_F, FWR_M, and FWR_H at the 120 kg stage (Figure 2B). The results showed that walnut kernel cake may promote adipose deposition and improve pork quality.

### 3.2. Walnut Kernel Cake Causes Significant Alteration in the Adipose Transcriptome

In order to study the effect of walnut kernel cake on the pig adipose transcriptome, we collected adipose tissue from pigs at two stages of 80 kg and 120 kg for transcriptome sequencing. It was found that walnut kernel cake had a significant effect on the pig adipose transcriptome at both stages (Figure 3A,B). Using the criteria of *p* < 0.05 and fold change > 2, we screened 378 and 687 differentially expressed genes (DEGs) at the two stages, respectively, among which there were 153 and 225 up-regulated and down-regulated genes for the feed B group (walnut kernel cake diet) at the 80 kg stage, and 310 and 377 up-regulated and down-regulated genes for the feed B group (walnut kernel cake diet) at the 120 kg stage (Figure 3C). Furthermore, we found that there were 44 shared DEGs at the two stages of 80 kg and 120 kg, of which 14 were shared up-regulated DEGs (*ANGPTL8*, *CCNP*, *ETV4*, *MESP1*, *MLANA*, *NUDT7*, *TRIB3*, etc.) and 12 were shared down-regulated DEGs (*ABCC11*, *ACTG2*, *DES*, *ENPP3*, *MMRN1*, *TBX1*, etc.) (Figure 3D). In addition, there were 18 DEGs with opposite expression trends in the two stages.

### 3.3. GO Enrichment Analysis Reveals Adipose Deposition-Related Biological Processes and Genes

To study the function of DEGs at two body weight stages of 80 kg and 120 kg, we performed GO term enrichment analysis using DEGs. At the 80 kg body weight stage, we obtained 74 significantly enriched GO terms (*p* < 0.05), and all these GO terms belonged to Biological Processes (BP). The word cloud annotation shows that these GO terms are mainly enriched in cell–cell signaling, organic acid metabolic process, etc. (Figure 4A). We classified the top 30 significantly enriched GO terms, which can be divided into five categories: negative behavior cycle activity, generation glycogen metabolites energy, diterpenoid fat-soluble retinoid hormone, blood tube diameter circulatory and endoderm endodermal formation development (Figure 4B). We further listed the significant enrichment of the top 20 GO terms and found that negative regulation of the Wnt signaling pathway related to adipose deposition was significantly enriched. Moreover, sugar and energy metabolism-related terms such as regulation of the glycogen metabolic process, regulation of the polysaccharide metabolic process and regulation of generation of the precursor metabolites and energy were significantly enriched (Figure 4C). This reveals that, at this stage, walnut kernel cake may affect the sugar and energy metabolism and further affect adipose deposition. To study the relationship between GO terms and genes, we further displayed the top 10 GO terms and the genes involved. We found that the genes *SOSTDC1*, *GRB10* and *EGR1* were significantly enriched in the negative regulation of the Wnt signaling pathway, and *PPP1R3B* and *PHLDA2* were significantly enriched in the regulation of the glycogen metabolic process (Figure 4D), suggesting that these genes may have an important role in adipose deposition.

At the 120 kg body weight stage, we obtained 546 significantly enriched GO terms (*p* < 0.05), all of which belonged to BP. The word cloud annotation shows that these GO terms are mainly enriched in epithelium development, organic acid metabolic process, and regulation of cell differentiation (Figure 4E). We classified the top 30 significantly enriched GO terms, which can be divided into five categories: growth multicellular development organism, ameboidal-type cell motility migration, epithelial tissue branching, MAPK cascade signal communication and purine nucleotide bisphosphate biosynthetic (Figure 4F). We further showed the significant enrichment of the top 20 GO terms, and found that the adipose-related MAPK cascade term was significantly enriched. Furthermore, multiple epithelium-related terms were significantly enriched (Figure 4G). To study the relationship between GO terms and genes, we further displayed the top 10 GO terms and the genes involved. We found that adipose deposition-related genes such as *WNT2*, *WNT11* and *SOX9* were significantly up-regulated (Figure 4H). From the number of enriched terms, it can be seen that the number of significantly enriched terms at the 120 kg body weight stage was significantly higher than that at the 80 kg body weight stage, indicating that walnut kernel cake might have a greater impact on pigs at the later stage of growth than at the early stage. 

### 3.4. KEGG Enrichment Analysis Reveals Adipose Deposition-Related Pathways and Genes

We investigated the KEGG enrichment pathways of DEGs at two body weight stages of 80 kg and 120 kg. At the 80 kg body weight stage, we obtained 41 significantly enriched pathways (*p* < 0.05). The word cloud annotation shows that these pathways are mainly enriched in the calcium signaling pathway, ECM–receptor interaction, MAPK signaling pathway, and insulin secretion, and the latter three of which are related to adipose deposition (Figure 5A). We classified the top 30 significantly enriched pathways, which can be divided into five categories: African AGE-RAGE secretion action, Amphetamine Calcium Circadian ataxia, Axon Butanoate Cytokine–cytokine guidance, Neuroactive Nicotine ligand–receptor addiction and ECM–receptor Focal Human adhesion (Figure 5B). We further showed the significantly enriched pathways of the top 20 and found some adipose deposition-related pathways such as insulin secretion, ECM–receptor interaction, and Wnt signaling pathway were significantly enriched (Figure 5C). To study the relationship between pathways and genes, we further showed the top 10 pathways and the genes involved (Figure 5D).

At the 120 kg body weight stage, we obtained 30 significantly enriched pathways (*p* < 0.05). The word cloud annotation shows that these pathways are mainly enriched in the calcium signaling pathway, MAPK signaling pathway, PPAR signaling pathway, and insulin signaling pathway (Figure 5E). We classified the top 30 significantly enriched pathways, which can be divided into five categories: acute adhesion B6 biosynthesis, ECM–receptor human digestion infection and carbon citrate cycle (TCA cycle) (Figure 5F). We further showed the significantly enriched pathways of the top 20, and also found that adipose deposition-related pathways such as the PPAR signaling pathway and Wnt signaling pathway were significantly enriched (Figure 5G). To study the relationship between pathways and genes, we further displayed the top 10 pathways and the genes involved, and found that the genes in the PPAR signaling pathway were significantly up-regulated, such as *PPARD*, *PLINNN5*, *CYP4A24*, *ACSL1*, *FABP3*, *FABP7*, *ME1*, etc. (Figure 5H), indicating that walnut kernel cake may regulate adipose deposition by up-regulating the PPAR signaling pathway.

Furthermore, we found eight common significantly enriched pathways in the two stages: melanogenesis, neuroactive ligand–receptor interaction, calcium signaling pathway, Wnt signaling pathway, PI3K–Akt signaling pathway, Human papillomavirus infection, ECM–receptor interaction, and retinol metabolism. Among them, the Wnt signaling pathway, PI3K–Akt signaling pathway and ECM–receptor interaction are related to adipose deposition, indicating that walnut kernel cake can further affect adipose deposition by significantly affecting these two pathways. 

### 3.5. GSEA Reveals That the PPAR Signaling Pathway Was Activated by Walnut Kernel Cake

To investigate gene set enrichment at both stages, we performed GSEA with all genes. At the 80 kg body weight stage, we obtained 41 and 11 significantly enriched GO terms and KEGG pathways, respectively (|NES| > 1, *p* < 0.05, *p*.adjust < 0.25). These include 9 up-regulated GO terms and 2 up-regulated KEGG pathways (|NES| > 1, *p* < 0.05, *p*.adjust < 0.25), 32 down-regulated GO terms and 9 down-regulated KEGG pathways (|NES| > 1, *p* < 0.05, *p*.adjust < 0.25). Among the top 10 GO terms, only the ribonucleoprotein complex was up-regulated in the walnut kernel cake additional group (Figure 6A). In the top 10 KEGG pathways, only the ribosome and synaptic vesicle cycles were up-regulated in the walnut kernel cake additional group (Figure 6B). At the 120 kg body weight stage, we obtained 205 and 38 significantly enriched GO terms and KEGG pathways, respectively (|NES| > 1, *p* < 0.05, *p*.adjust < 0.25). These include 145 up-regulated GO terms and 22 up-regulated KEGG pathways (|NES| > 1, *p* < 0.05, *p*.adjust < 0.25), 60 down-regulated GO terms and 16 down-regulated KEGG pathways (|NES| > 1, *p* < 0.05, *p*.adjust < 0.25). All the top 10 GO terms were up-regulated in the walnut kernel cake additional group, and many terms were related to mitochondria (Figure 6C). Among the top 10 KEGG pathways, the PPAR signaling pathway was up-regulated in the walnut kernel cake additional group (Figure 6D). The genes *ACSL1*, *CYP4A24*, *FABP3*, *FABP7*, *ME1*, *PLIN5*, *PPARD* and *RXRG* involved in this pathway are all related to adipose deposition, and all of them are significantly up-regulated (Figure 6E). This shows that walnut kernel cake promotes adipose deposition in pigs by activating the PPAR signaling pathway and the related genes.

### 3.6. WGCNA Identifies Two Modules Significantly Associated with Adipose Traits

To study gene sets associated with traits, we performed WGCNA and obtained 21 modules (Figure 7A). Using the hub genes in each module, we performed cluster analysis on these 21 modules (Figure 7B). Further, we used the expression level of genes in each module to perform a correlation analysis with the 11 adipose-related traits. The results showed that the module tan was highly correlated with adipose traits, especially with BF and AFR (Figure 7C). The correlation coefficients between the module tan and AFR, BF_A, BF_B, BF_C, and BF_Avg were above 0.5 (*p* < 0.05). It indicated that walnut kernel cake mainly affected traits such as BF and AF through the gene sets of this module.

### 3.7. Correlation Analysis Identifies Adipose Deposition-Related Genes

In order to study the genes that significantly affect adipose-related traits, we extracted the 182 genes in the tan module and performed correlation analysis with the 11 traits. Among the 2002 gene–trait pairs, 432 (21.58%) were significantly correlated (*p* < 0.05, |*r*| > 0.576), of which 169 had |*r*| > 0.7, involving 88 genes, such as *ABTB2*, *ADAMTS18*, *AHSP*, *ALAS2* and *ATP6V1G2* (Figure 8A). Among them, *ALAS2*, *DUSP4*, *ENSSSCG00000007978*, *ENSSSCG00000036334*, *HK2*, *KCNS3* and *LRATD1* are the DEGs in the 80 kg stage, and a total of 17 genes such as *ABTB2*, *DUSP4*, *ENSSSCG00000001458*, *ENSSSCG00000025367*, *ENSSSCG00000036334*, *EVPL*, *HK2*, *HMCN1*, *HSPA12A*, *KCNH3*, *KIF17*, *LRATD1*, *NR1I2*, *RGS7*, *SMAD7*, *TNN* and *U6* are the DEGs in the 120 kg stage.

At the same time, we also used the DEGs of the two stages of 80 kg and 120 kg to conduct correlation analysis with these 11 traits and found that there were 147 pairs and 735 pairs of significant correlations in the two stages (*p* < 0.05, |*r*| > 0.576). Among them, there were 35 gene–trait pairs with |*r*| > 0.7 in the 80 kg stage, involving 30 genes, in which three genes such as *ALAS2*, *ADRB1* and *ENSSSCG00000016467* had correlation coefficients greater than 0.8 (Figure 8B). In the 120 kg stage, there were 185 gene–trait pairs involving 154 genes with |*r*| > 0.7. Among them, 35 gene–trait pairs with 30 genes had correlation coefficients greater than 0.8 (Figure 8C).

Notably, genes in 96% (24/25) of the negatively correlated gene–trait pairs were significantly down-regulated, and genes in 80% (8/10) of the positively correlated gene–trait pairs were significantly up-regulated. Namely, 91.43% (32/35) of the gene–trait pairs promoted adipose deposition, involving 90% (27/30) of the genes. Among the 32 gene–trait pairs that promote adipose deposition, 25 were related to FWR, involving the genes *ADRB1*, *CCDC173*, *CDC45*, *DCLK3*, *DUSP10*, *ENSSSCG00000001081*, *ENSSSCG00000001458*, *ENSSSCG00000013869*, *ENSSSCG00000016467*, *ENSSSCG00000038037*, *FBP2*, *GATM*, *GPR18*, *LEP*, *MAP3K15*, *NECTIN1*, *PLA2G10*, *RNASE4*, *SLC25A45*, and *ZGRF1*. There were six gene–trait pairs related to backfat thickness, such as *CKB*, *EEF1A2*, *ENSSSCG00000033248*, *ENSSSCG00000040134*, *PER2* and *PTGES*. This further explained that walnut kernel cake mainly affected adipose-related traits such as FWR and BF at the 120 kg stage.

### 3.8. Single-Cell Transcriptome Analysis Reveals Key Genes for Adipose Deposition

In order to explore the regulatory mechanism of walnut kernel cake on adipose deposition at the single-cell level, we used the published single-cell transcriptome data of porcine adipose tissue to conduct an in-depth analysis of genes significantly related to adipose traits. We initially obtained data with 10,890 cells covering 15,490 genes. After quality control, we finally retained 9941 cells, and after further dimensionality reduction, a total of 17 clusters were obtained (Figure 9A). Combining CellMarker, PanglaoDB and the classic adipose tissue marker genes in the literature, we annotated these 17 clusters as seven cell types, including adipocyte progenitor/stem cells (APSCs), cycling cells, endothelial cells, lymphatic endothelial cells, macrophages, NK/T cells and smooth muscle cells (Figure 9B). Figure 9C shows the marker genes we used to annotate cell types (Figure 9C). Next, we investigated gene expression that was significantly associated with the traits in each cell type. The results showed that these genes were mainly expressed in non-immune cell types such as APSCs, cycling cells, endothelial cells and smooth muscle cells. Among them, *CPXM2*, *FMOD*, *SMOC2*, *VIPR2*, *CD248* and *PTGES* were specifically expressed in APSCs. *ZNF367* and *ENSSSCG00000005481* genes were specifically expressed in cycling cells. *PPP1R3B*, *SULT1C4*, *RNF125*, *SELP* and *VEGFC* genes were specifically expressed in endothelial cells (Figure 9D–F). The above genes, especially the cell type-specific expression genes of APSCs, may play an important role in regulating adipogenesis and deposition. In addition, we found that *EGR1*, *FOS* and *TXNIP* genes were relatively highly conserved in various cell types and may also play an important role in regulating adipose deposition.

## 4. Discussion

In the field of pig breeding, research has been dedicated to finding efficient and economical feed alternatives that can improve meat quality. Walnut cake is a by-product of walnut processing and is usually used as a feed additive [14,15]. It is rich in protein, fat, fiber, minerals, vitamins, phytic acid and polyphenolic compounds, which may play an important role in regulating the nutritional needs and growth performance of animals [29,30]. However, there are few reports on the effect of walnut kernel cake on adipose deposition in pigs. In this study, substituting walnut kernel cake for part of the soybean protein in feed was found to have an effect on adipose deposition in large Diqing Tibetan pigs and altered the gene expression profile of adipose tissue.

In this study, walnut kernel cake significantly promoted adipose deposition and improved pork quality in pigs. Specifically, it significantly increased the caul fat rate of pigs at the 80 kg stage, and significantly increased the backfat thickness and FWR of pigs at the 120 kg stage. Some studies have shown that walnut functional food or walnut meal will not cause obesity or adipose deposition [31,32,33,34]. However, the walnut kernel cake in this study caused adipose deposition in pigs, probably because the walnut kernel cake contained more lipids compared to the partially replaced soybean meal, which is consistent with the experimental results of Untea et al. in chickens [14,15]. The effects of fiber, minerals, vitamins, phytic acid and polyphenolic compounds in walnut cake on pig fat deposition and their regulatory mechanisms need to be further explored.

Walnut kernel cake also significantly altered the transcriptome level of porcine adipose tissue. Among them, the number of DEGs in the 120 kg stage was 1.82 times that in the 80 kg stage, indicating that the walnut kernel cake played an important role in the adipose deposition of pigs, especially in the late fattening period. We found a total of 26 shared genes in the two stages, among which *ANGPTL8* is an adipocytokine known to play an important regulatory role in fat metabolism. The study found that *ANGPTL8* can promote the maturation of adipocytes and the release of fatty acids, while regulating insulin sensitivity and glucose metabolism [35,36,37,38]. *ETV4* may play a role in the regulation of adipocyte differentiation and metabolism [39,40]. *TRIB3* is involved in regulating cell proliferation, differentiation and apoptosis [41]. In adipocytes, *TRIB3* may interact with the insulin signaling pathway to affect fat metabolism and insulin sensitivity [42]. These genes may play an important regulatory role in adipose deposition.

The functional enrichment results of differentially expressed genes showed that multiple fat-related pathways were significantly enriched, including the PPAR signaling pathway, insulin signaling pathway, PI3K-Akt signaling pathway, Wnt signaling pathway, MAPK signaling pathway, etc. These pathways have key regulatory roles in adipocyte differentiation, proliferation and fatty acid synthesis, which can promote adipocyte maturation, increase adipocyte number and adipose deposition [43,44,45,46,47,48]. GSEA analysis showed that the walnut kernel cake activated the PPAR signaling pathway at the 120 kg stage. It plays a key regulatory role in adipocyte differentiation and fatty acid synthesis. The PPAR signaling pathway can promote the differentiation process of adipose stem cells into adipocytes, and promote the differentiation and maturation of adipocytes [49,50]. In addition, it can also promote fatty acid synthesis and triacylglycerol synthesis, thereby increasing adipose deposition and storage [51].

The PPAR signaling pathway is very important for the regulation of insulin sensitivity. Insulin is an important metabolic hormone that promotes glucose uptake and utilization and inhibits fatty acid release. Activating the PPAR signaling pathway can improve the sensitivity of cells to insulin, and promote the uptake and metabolism of glucose by adipose cells, thereby reducing the release of fatty acids and inhibiting the decomposition of adipose tissue [52,53]. This pathway may play an important regulatory role in the process of walnut kernel cake promoting adipose deposition. All DEGs involved in this pathway, such as *ACSL1*, *CYP4A24*, *FABP3*, *FABP7*, *ME1*, *PLIN5*, *PPARD* and *RXRG,* were related to adipose deposition, and were significantly up-regulated in the walnut kernel cake-supplemented group. In addition, we also found mitochondrial and energy metabolism-related pathways and the role of these pathways in adipose deposition in this study needs further study.

We screened some DEGs significantly associated with adipose traits through WGCNA and correlation analysis. In the 120 kg stage, we found many genes highly correlated with adipose traits (|*r*| > 0.8, *p* < 0.05). Among them, 96% of DEGs negatively correlated with adipose traits were down-regulated, and 80% of DEGs positively correlated with adipose traits were up-regulated. The above results further indicated that walnut kernel cake feeding can promote adipose deposition in pigs. Among them, *Per2* and *Cry2* interact to promote adipogenesis by inhibiting the Wnt signaling pathway in mice [54]. The increase in *PTGES* may promote the synthesis of prostaglandin E2, which in turn regulates the metabolic activity of adipocytes. Prostaglandin E2 is considered to be a biologically active substance that can promote adipocyte proliferation and fat synthesis. It can promote the differentiation and proliferation of adipocytes and increase the synthesis and deposition of fat by activating *PGE2* receptors in adipocytes [55]. The up-regulation of these genes may play an important regulatory role in the process of walnut kernel cake promoting adipose deposition. In addition, we also found some other DEGs that regulated adipose deposition, such as CKB and LEP [56,57]. How the above genes and pathways interact to promote adipose deposition in the present study needs further exploration.

There are few studies exploring adipose deposition at the single-cell level. This study further explored the expression of these key genes in different adipose tissue cell types using single-cell transcriptome data. It was found that *CPXM2*, *FMOD*, *SMOC2*, *VIPR2*, *CD248*, *PTGES* and other genes were specifically expressed in APSCs. APSCs are adipocyte precursors and stem cells that can differentiate into mature adipocytes and promote adipose deposition [58]. Among them, *CD248* and *PTGES* were significantly up-regulated at the 120 kg stage, and these two genes were highly positively correlated with backfat thickness, indicating that these two genes may mainly act on APSCs to promote the differentiation of adipose precursor stem cells and then promote the formation of adipocytes and adipose deposition.

The limitation of this study is that the sample size used is slightly smaller. Although we have used more stringent standards to screen differentially expressed genes and analyze trait differences, there may still be some bias in the results. Therefore, we will conduct functional experimental verification of some key genes in the future. A sufficient number of samples will be used for experiments in future studies to ensure the credibility and representativeness of the data.

## 5. Conclusions

In the present study, we found that incorporating walnut kernel cake into feed can efficiently replace a portion of soybean cake, offering pigs high-quality protein and fostering adipose deposition. This process is regulated by the transcriptome and single-cell transcriptome. Although we identified some genes and pathways related to adipose deposition, adipose deposition is a complex biological process involving the combined action of multiple genes and regulatory networks. An in-depth study of the functions of these genes and pathways in adipose metabolism and deposition will help to better understand their roles in animal breeding and adipose-metabolism-related diseases in humans. This study provides a theoretical basis for feed protein replacement, pig genetic breeding, and meat quality improvement and also provides reference materials for the study of human fat metabolism and related diseases.

## Figures and Tables

**Figure 1 genes-15-00667-f001:**
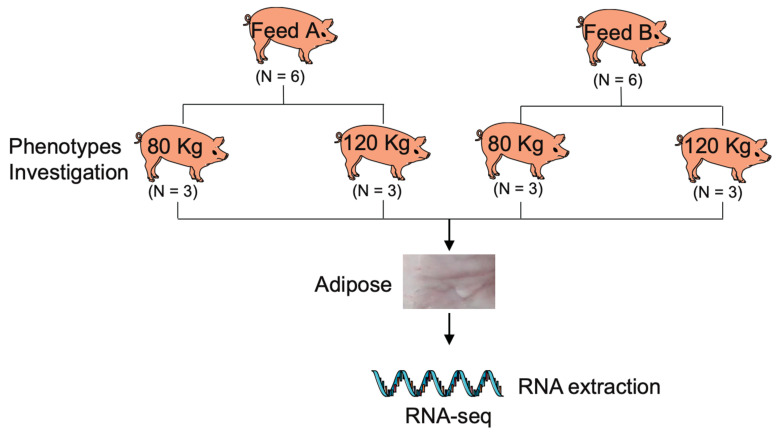
Experimental design flowchart. Phenotypes include 11 traits such as caul fat rate (CFR), abdominal fat rate (AFR), backfat thickness at three positions (BF_A, BF_B, BF_C), average backfat thickness (BF_Avg), backfat thickness between the 6th and 7th ribs (BF_67), fat weight rate at three positions (FWR_F, FWR_M, FWR_H), and total fat weight rate (FWR_T).

**Figure 2 genes-15-00667-f002:**
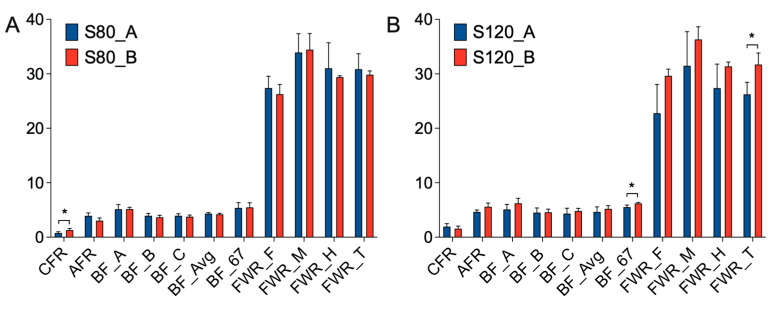
Trait determination at the 80 and 120 kg body weight stages. (**A**,**B**) Phenotype investigation at the 80 kg (**A**) and 120 kg (**B**) body weight stages, including 11 traits such as caul fat rate (CFR), abdominal fat rate (AFR), backfat thickness at three positions (BF_A, BF_B, BF_C), average backfat thickness (BF_Avg), backfat thickness between the 6th and 7th ribs (BF_67), fat weight rate at three positions (FWR_F, FWR_M, FWR_H), and total fat weight rate (FWR_T). The y-axis of CFR, AFR, FWR_F, FWR_M, FWR_H and FWR_T represents percentage, and the unit is %. The y-axis of BF_A, BF_B, BF_C, BF_Avg and BF_67 represents thickness, and the unit is centimeter (cm). BF80_A: Pigs fed with Feed A at 80 kg stage. BF80_B: Pigs fed with Feed B at 80 kg stage. BF120_A: Pigs fed with Feed A at 120 kg stage. BF120_B: Pigs fed with Feed B at 120 kg stage. The error bars represent standard deviation. * *p* < 0.05.

**Figure 3 genes-15-00667-f003:**
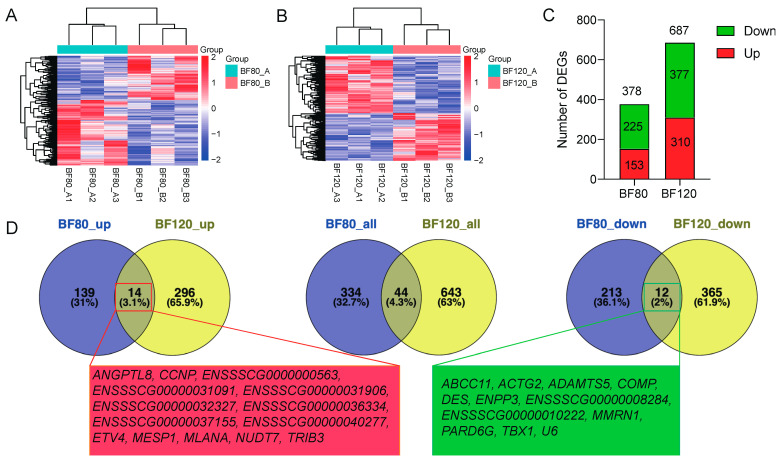
Differential expression analysis of adipose transcriptome at the 80 kg and 120 kg body weight stages. (**A**) Heatmap of differentially expressed genes (DEGs) at 80 kg stage, which showed the feed effect on the group of pigs at the 80 kg stage. (**B**) Heatmap of DEGs at 120 kg stage, which showed the feed effect on the group of pigs at the 120 kg stage. (**C**) Statistics on the number of DEGs. (**D**) Venn diagram of DEGs at the two stages. The left is a Venn diagram of up-regulated DEGs. The middle is a Venn diagram of all DEGs. The right is the Venn diagram of down-regulated DEGs. The genes in the red and green boxes below are the up-regulated and down-regulated DEGs shared by the two stages, respectively. BF80_A: pigs fed with Feed A at 80 kg stage. BF80_B: pigs fed with Feed B at 80 kg stage. BF120_A: pigs fed with Feed A at 120 kg stage. BF120_B: pigs fed with Feed B at 120 kg stage. BF80: pigs at 80 kg stage. BF120: pigs at 120 kg stage.

**Figure 4 genes-15-00667-f004:**
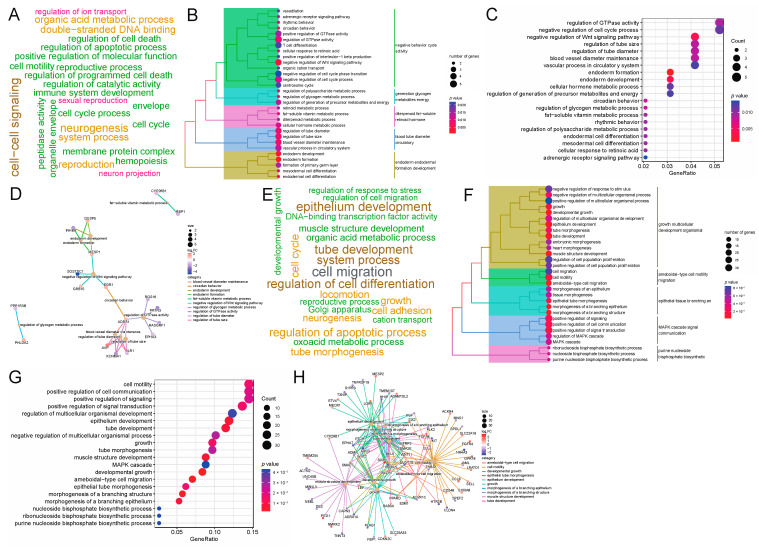
GO term enrichment analysis of DEGs at the 80 kg and 120 kg body weight stages. (**A**) GO term word cloud annotation at the 80 kg body weight stage. As the font size of a word increases, its frequency in the terms also increases. The colors of different entries are randomly assigned. (**B**) Taxonomic summary of the GO terms at the 80 kg body weight stage. (**C**) Top 20 GO terms at the 80 kg body weight stage. (**D**) Top 10 GO terms and their enriched gene interactions at the 80 kg body weight stage. (**E**) GO term word cloud annotation at the 120 kg body weight stage. As the font size of a word increases, its frequency in the terms also increases. The colors of different entries are randomly assigned. (**F**) Taxonomic summary of the GO terms at the 120 kg body weight stage. (**G**) Top 20 GO terms at the 120 kg body weight stage. (**H**) Top 10 GO terms and their enriched gene interactions at the 120 kg body weight stage.

**Figure 5 genes-15-00667-f005:**
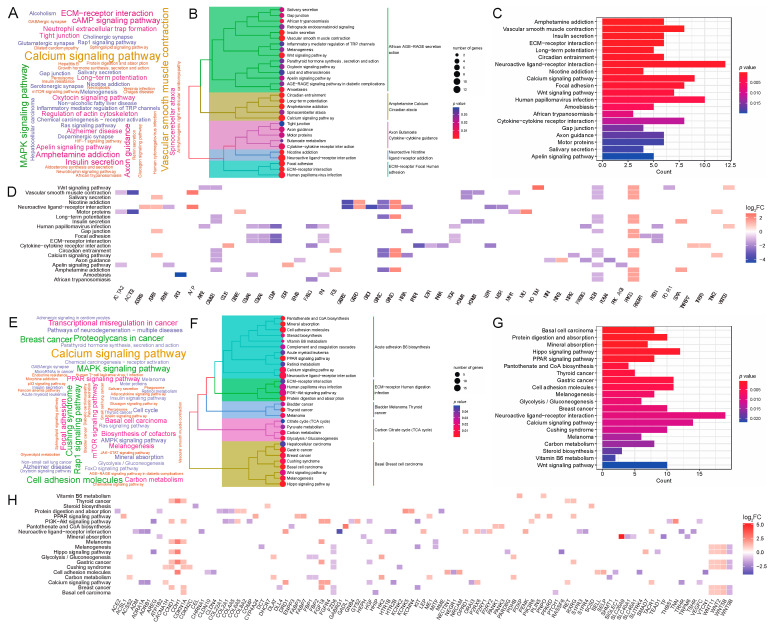
KEGG enrichment analysis of DEGs at the 80 kg and 120 kg body weight stages. (**A**) KEGG pathway word cloud annotation at the 80 kg body weight stage. As the font size of a word increases, its frequency in the terms also increases. The colors of different entries are randomly assigned. (**B**) Taxonomic summary of the pathways at the 80 kg body weight stage. (**C**) Top 20 pathways at the 80 kg body weight stage. (**D**) Top 10 pathways and their enriched gene interactions at the 80 kg body weight stage. (**E**) KEGG pathway word cloud annotation at the 120 kg body weight stage. As the font size of a word increases, its frequency in the terms also increases. The colors of different entries are randomly assigned. (**F**) Taxonomic summary of the pathways at the 120 kg body weight stage. (**G**) Top 20 pathways at the 120 kg body weight stage. (**H**) Top 10 pathways and their enriched gene interactions at the 120 kg body weight stage. Red and blue squares indicate gene up- and down-regulation, respectively.

**Figure 6 genes-15-00667-f006:**
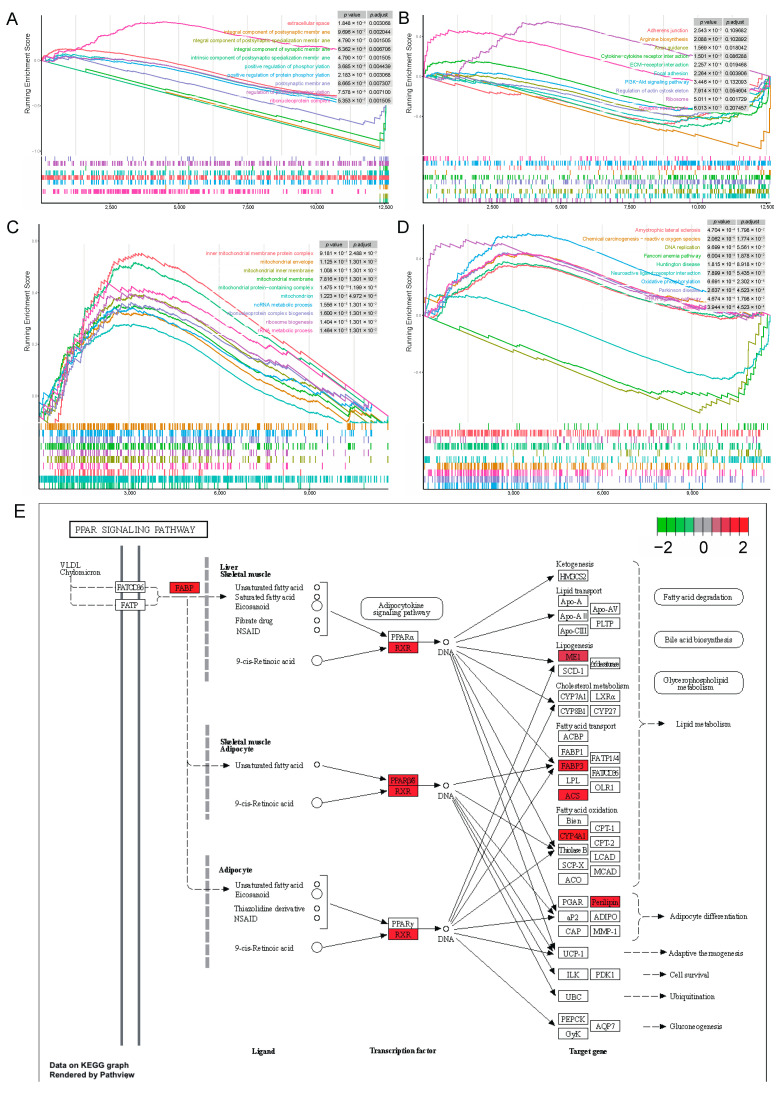
Gene set enrichment analysis of the 80 kg and 120 kg body weight stages. (**A**) GSEA of GO term at the 80 kg body weight stage. (**B**) GSEA of KEGG pathway at the 80 kg body weight stage. (**C**) GSEA of GO term at the 120 kg body weight stage. (**D**) GSEA of KEGG pathway at the 120 kg body weight stage. (**E**) PPAR signaling pathway and its related DEGs at the 120 kg body weight stage.

**Figure 7 genes-15-00667-f007:**
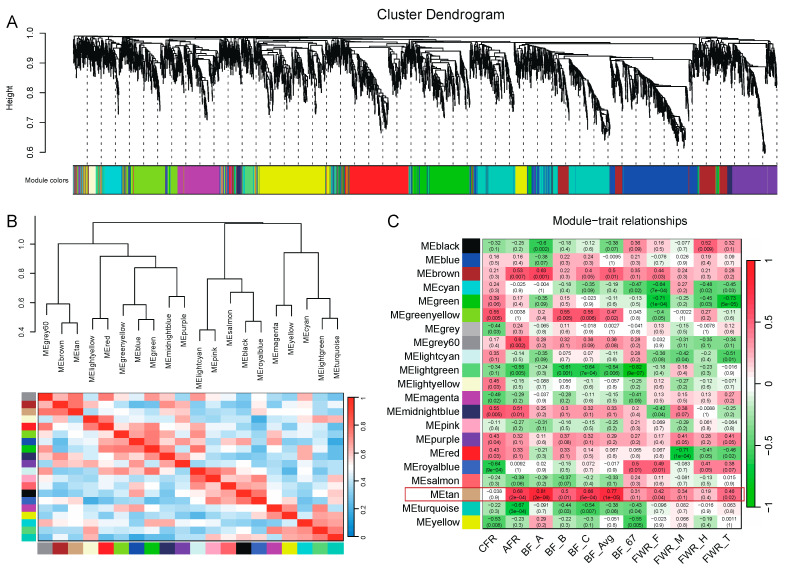
Construction of co-expression modules based on genes in all samples. (**A**) Cluster dendrogram of genes. Each branch represents one gene, and every color below represents one co-expression module. (**B**) Hierarchical clustering and heatmap of modules. Different colors at the bottom represent different modules. The grey module contains genes that are not significantly correlated with genes in other modules and is removed here. (**C**) Heatmap of the correlation between module eigengenes and adipose-related traits. The grey60 and tan modules were the most positively correlated with traits.

**Figure 8 genes-15-00667-f008:**
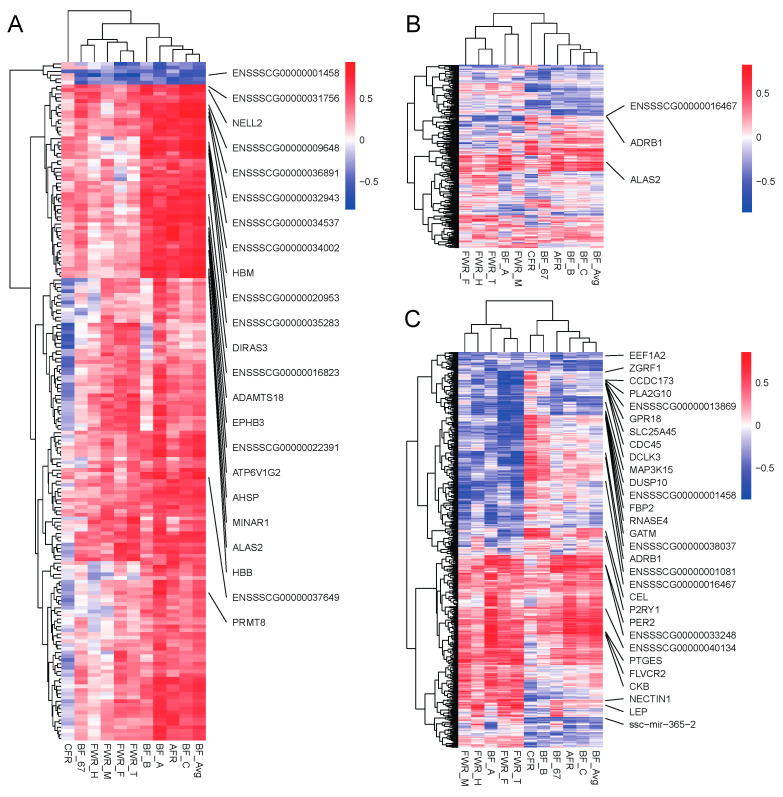
Correlation analysis of genes and adipose-related traits. (**A**) Heatmap of the correlation coefficients between genes and adipose-related traits in the tan module. (**B**) Heatmap of correlation coefficients between DEGs and adipose-related traits at the 80 kg body weight stage. (**C**) Heatmap of correlation coefficients between DEGs and adipose-related traits at the 120 kg body weight stage. Each row represents a gene, and each column represents a trait. Gene names with correlation coefficients greater than 0.8 for traits are displayed.

**Figure 9 genes-15-00667-f009:**
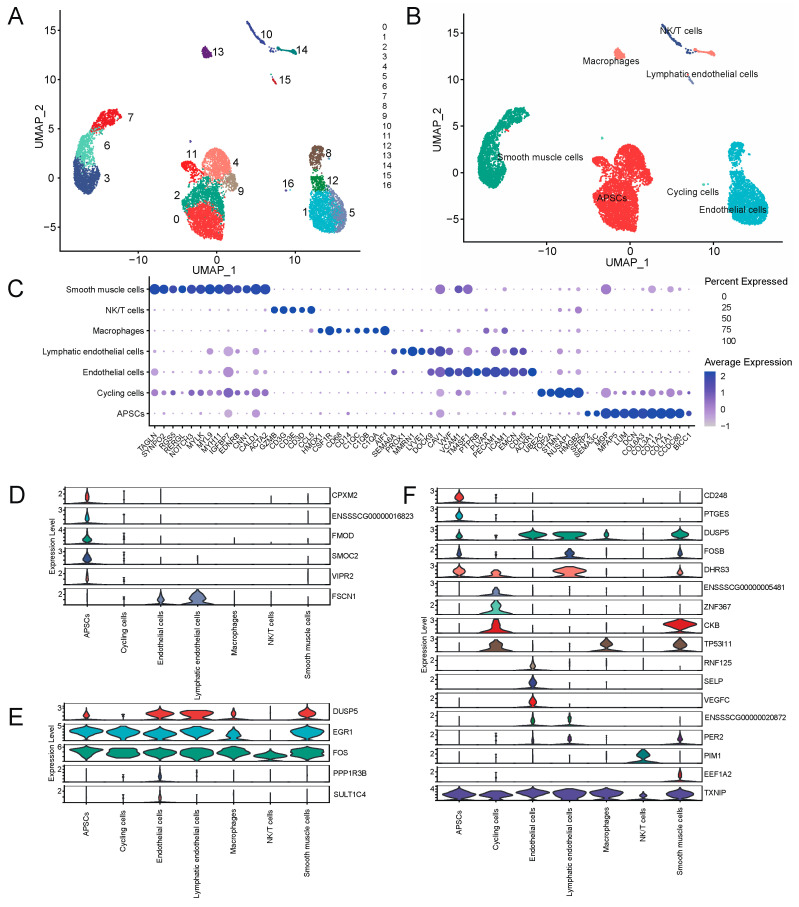
Single-cell transcriptome analysis of porcine adipose tissue. (**A**) UMAP plot of cell clusters in pig adipose tissue. (**B**) UMAP plot of cell types in adipose tissue. (**C**) Dotplot of marker genes in different cell types. (**D**) Violin plots of genes that are significantly associated with adipose-related traits in the grey60 and tan modules. (**E**) Violin plot of DEGs that are significantly associated with adipose-related traits at the 80 kg body weight stage. (**F**) Violin plot of DEGs that are significantly associated with adipose-related traits at the 120 kg body weight stage.

**Table 1 genes-15-00667-t001:** The nutritional value of walnut cake.

Nutritional Value/%	Value *	SE *
Water	8.1320419	0.0290737
Crude protein	22.61978519	0.450726075
Ca	0.4539975	0.00827276
TP	0.5211185	0.3548774
Ash	2.7819314	0.049862
Crude fiber	33.7111167	0.5357425
Neutral detergent fiber	49.2587963	0.8416535
Acid detergent fiber	33.1360225	0.7875224
Crude fat	8.1654504	0.1081171

* The nutritional value of walnut cake; we measured three replicates and finally took the average value. SE represents the standard error of the three replicates.

**Table 2 genes-15-00667-t002:** Composition and nutritional value of the experimental diets.

Ingredients/%	8–15 kg		15–30 kg		30–60 kg		60–120 kg	
	Feed A *	Feed B *	Feed A *	Feed B *	Feed A *	Feed B *	Feed A *	Feed B *
Corn	63.61	61.7	44.9	40.4	47.7	43.3	50.4	45.7
Soybean meal	23.54	21.4	13.39	9.7	10.7	7.3	8	3.9
Wheat bran	3	3	36.3	39.4	36.3	39	36.3	40
Stone powder	1	1	1	1	1	1	1	1
Soybean oil	1.4	0.4						
Fish meal	3	3						
Walnut cake		5		5		5		5
Nacl	0.3	0.3	0.3	0.3	0.3	0.3	0.3	0.3
Lys (78.5%)	0.15	0.2	0.11	0.2		0.1		0.1
Premix	4	4	4	4	4	4	4	4
Total	100	100	100	100	100	100	100	100
Nutritional value								
DE (MJ/kg)	13.60	13.60	11.70	11.77	11.72	11.76	11.72	11.76
CP	18.22	18.23	15.02	15.09	14.07	14.07	13.11	13.11
CF	2.62	4.15	3.87	5.45	3.75	5.36	3.64	5.25
Met + Cys	0.62	1.11	0.54	1.00	0.51	0.98	0.49	0.96
Lys	1.07	1.07	0.78	0.78	0.63	0.63	0.56	0.57
Ca	0.76	0.77	0.64	0.65	0.63	0.64	0.62	0.63
P	0.62	0.62	0.72	0.74	0.71	0.74	0.70	0.73

* The corresponding value below each growth stage is the feed formula for this stage. At the beginning of the experiment, Feed A and Feed B groups each contained 6 pigs. When the body weight of the pigs reached 80 kg, three pigs fed with Feed A and three pigs fed with Feed B were slaughtered and the backfat tissues were collected. 8–80 kg, N_FeedA_ = 6, N_FeedB_ = 6. When the body weight reached 120 kg, three pigs fed with Feed A and three pigs fed with Feed B were slaughtered and the backfat tissues were collected. 80–120 kg, N_FeedA_ = 3, N_FeedB_ = 3.

## Data Availability

The RNA-seq datasets supporting the conclusions of this article are available in the National Center for Biotechnology Information (NCBI) database. The corresponding accession number is PRJNA1003490.

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
