# Peer review of "Transcriptomic Analyses Reveal the Effects of Walnut Kernel Cake on Adipose Deposition in Pigs"

_genes, 2024, doi:10.3390/genes15060667_

Round 1

Reviewer 1 Report

Comments and Suggestions for Authors

I believe there should be a better introduction to the subject and justify why the use of walnut cake in pig diets is rarely reported.

In the paragraph stating the objectives, there is a partial conclusion of the study which should not be referred to in this section.

In material and methods: the sample size is too small. Why were only 12 animals used? This needs to be justified. What effect does the sample size have on the inference made?

Too many traits are studied in relation to the number of animals used in the reported experiment.

The genomic part is adequately described, the statistical analyses are described in a very cursory manner, this can be improved.

Discussion: In general the discussion is clearly and adequately conducted. I think the authors do an excellent job in describing and discussing their results, but the main concern is in the experimental setup. There are too few animals to make adequate inference from the information that is generated.

Conclusions should be improved by highlighting the main findings, but not in the form of results. The second part of the conclusions seems to me to be in a better form.

Author Response

Reviewers' Comments:

Reviewer #1:

I believe there should be a better introduction to the subject and justify why the use of walnut cake in pig diets is rarely reported.

Response: Thanks for your suggestion. We rewrote the introduction as following:

“As a nutritious food, research on walnuts in human diet has always attracted much attention [1, 2]. Multiple studies have linked walnut intake to cardiovascular health. Walnuts are rich in unsaturated fatty acids, phytosterols and fiber, which can help lower cholesterol levels, improve blood lipid metabolism, and reduce the risk of cardiovascular disease [3-5]. Walnuts are rich in antioxidants, such as vitamin E, polyphenols and an-tioxidant enzymes, which help neutralize free radicals, reduce oxidative stress damage, and improve the body's antioxidant capacity [6]. Some studies suggest that walnut consumption may help improve cognitive function and brain health [7]. Components such as omega-3 fatty acids and antioxidants in walnuts are thought to be beneficial to brain function [8, 9]. However, perhaps due to the problems of small global cultivation range and production, walnut and its ancillary products have been less studied in the field of livestock farming. China's walnut planting area and output rank first in the world, and it is also the country with the highest walnut consumption in the world [10]. Yunnan Province is the largest walnut producing area in China (accounting for 27.17% of the national walnut production), and Fengqing County is the main walnut producing area in Yunnan Province [11]. A large amount of walnut cake is produced here every year due to oil extraction. In line with a sustainable path, the concept of ‘waste to wealth’ leading to ‘green growth’ is a great opportunity to improve food security and is being adopted by many developed and developing countries [12].

In the pig breeding industry, seeking efficient and economical feed alternatives and improving meat quality have always been the focus of research. Adjustment of feed ingredients can not only reduce production costs, but also improve pig growth perfor-mance and meat production quality. Walnut cake is the solid residue left over from the oil extraction process from walnuts and is often used as feed or other agricultural purposes. It has attracted attention as a potential feed substitute because it is rich in protein and fat and has potential nutritional advantages [2]. Research on walnut kernel cake may provide new feed options for the pig industry and improve the health and nutrition of animal husbandry [13]. Partial replacement of soybean meal with walnut cake as a protein source has an impact on the meat quality of broiler breast meat [14, 15]. However, there are relatively few studies on the application of walnut cake in pig feed and its effect on pig fat deposition, especially on the Diqing Tibetan pig, a local breed in Yunnan Province, China [16]. Therefore, exploring the effect of walnut cake on pig fat deposition has important theoretical and application value.” Please refer to the lines 35-67.

Reference:

  1. Sandu-Balan Tabacariu, A.; Ifrim, I. L.; Patriciu, O. I.; Stefanescu, I. A.; Finaru, A. L., Walnut By-Products and Elderberry Extracts-Sustainable Alternatives for Human and Plant Health. Molecules 2024, 29, (2).
  2. Ni, Z. J.; Zhang, Y. G.; Chen, S. X.; Thakur, K.; Wang, S.; Zhang, J. G.; Shang, Y. F.; Wei, Z. J., Exploration of walnut components and their association with health effects. Crit Rev Food Sci Nutr 2022, 62, (19), 5113-5129.
  3. Petersen, K. S.; Chandra, M.; Chen See, J. R.; Leister, J.; Jafari, F.; Tindall, A.; Kris-Etherton, P. M.; Lamendella, R., Walnut consumption and gut microbial metabolism: Results of an exploratory analysis from a randomized, crossover, controlled-feeding study. Clin Nutr 2023, 42, (11), 2258-2269.
  4. Tepavcevic, S.; Romic, S.; Zec, M.; Culafic, T.; Stojiljkovic, M.; Ivkovic, T.; Pantelic, M.; Kostic, M.; Stanisic, J.; Koricanac, G., Effects of Walnut-Rich Diet on Cation-Handling Proteins in the Heart of Healthy and Metabolically Compromised Male Rats. J Med Food 2023, 26, (11), 849-857.
  5. Olas, B., The Cardioprotective Properties of Selected Nuts: Their Functional Ingredients and Molecular Mechanisms. Foods 2024, 13, (2).
  6. Fan, N.; Fusco, J. L.; Rosenberg, D. W., Antioxidant and Anti-Inflammatory Properties of Walnut Constituents: Focus on Personalized Cancer Prevention and the Microbiome. Antioxidants (Basel) 2023, 12, (5).
  7. Wu, W.; Niu, B.; Peng, L.; Chen, Q.; Chen, H.; Chen, H.; Xia, W.; Jin, L.; Simal‐Gandara, J.; Gao, H., Recent advances on the effect of nut consumption on cognitive improvement. Food Frontiers 2023, 4, (4), 1737-1746.
  8. Loong, S.; Barnes, S.; Gatto, N. M.; Chowdhury, S.; Lee, G. J., Omega-3 Fatty Acids, Cognition, and Brain Volume in Older Adults. Brain Sci 2023, 13, (9).
  9. Feng, J.; Zheng, Y.; Guo, M.; Ares, I.; Martinez, M.; Lopez-Torres, B.; Martinez-Larranaga, M. R.; Wang, X.; Anadon, A.; Martinez, M. A., Oxidative stress, the blood-brain barrier and neurodegenerative diseases: The critical beneficial role of dietary antioxidants. Acta Pharm Sin B 2023, 13, (10), 3988-4024.
  10. Ma, X.; Wang, W.; Zheng, C.; Liu, C.; Huang, Y.; Zhao, W.; Du, J., Quality Evaluation of Walnuts from Different Regions in China. Foods 2023, 12, (22).
  11. Zhou, X.; Peng, X.; Pei, H.; Chen, Y.; Meng, H.; Yuan, J.; Xing, H.; Wu, Y., An overview of walnuts application as a plant-based. Front Endocrinol (Lausanne) 2022, 13, 1083707.
  12. Sari, T.; Sirohi, R.; Krishania, M.; Bhoj, S.; Samtiya, M.; Duggal, M.; Kumar, D.; Badgujar, P. C., Critical overview of biorefinery approaches for valorization of protein rich tree nut oil industry by-product. Bioresource Technology 2022, 127775.
  13. Danilov, A.; Donică, I., The use of nut kernel cake in the feeding of young pigs. Scientific Papers. Series D. Animal Science 2022, 65, (2), 110-116.
  14. Untea, A. E.; Varzaru, I.; Saracila, M.; Panaite, T. D.; Oancea, A. G.; Vlaicu, P. A.; Grosu, I. A., Antioxidant Properties of Cranberry Leaves and Walnut Meal and Their Effect on Nutritional Quality and Oxidative Stability of Broiler Breast Meat. Antioxidants (Basel) 2023, 12, (5).
  15. Untea, A. E.; Turcu, R. P.; Saracila, M.; Vlaicu, P. A.; Panaite, T. D.; Oancea, A. G., Broiler meat fatty acids composition, lipid metabolism, and oxidative stability parameters as affected by cranberry leaves and walnut meal supplemented diets. Scientific Reports 2022, 12, (1), 21618.
  16. Cai, Y.; Quan, J.; Gao, C.; Ge, Q.; Jiao, T.; Guo, Y.; Zheng, W.; Zhao, S., Multiple domestication centers revealed by the geographical distribution of Chinese native pigs. Animals 2019, 9, (10), 709.

In the paragraph stating the objectives, there is a partial conclusion of the study which should not be referred to in this section.

Response: Thanks for your suggestion. We have deleted the sentence “In this study, we discovered that using walnut kernel cake as a substitute for soybean protein in feed could significantly alter adipose transcriptome levels, increase adipose deposition, and improve pork quality.” and added the sentences “We initially investigated the effects of walnut kernel cake on adipose-related traits at different growth and developmental stages of pigs. Subsequently, we dissected the molecular mechanisms underlying the alterations of adipose-related traits induced by walnut kernel cake at the transcriptome level. Finally, we further elucidated changes in adipocyte types and their marker genes at the single-cell transcriptome level during this process.” in the revised manuscript. Please refer to the lines 70-75.

In material and methods: the sample size is too small. Why were only 12 animals used? This needs to be justified. What effect does the sample size have on the inference made?

Response: Thank you for your comment. We understand your concerns about experimental reproducibility. In our study, we chose a design using at least 3 biological replicates per treatment, a decision made taking into account resources, cost, and experimental feasibility. Although the number of replications is relatively small, we adopted strict control and statistical methods in the experimental design and data analysis stages to ensure that our results have a certain degree of credibility and representativeness. We will also review your suggestions and consider increasing the number of replicates in future studies to further enhance our experimental design. We also addressed the limitations of this study at the end of the “Discussion” section as following: “The limitation of this study is that the sample size used is slightly smaller. Although we have used more stringent standards to screen differentially expressed genes and analyze trait differences, there may still be some bias in the results. Therefore, we will conduct functional experimental verification of some key genes in the future. A sufficient number of samples will be used for experiments in future studies to ensure the credibility and representativeness of the data”. Please refer to the lines 551-556.

Too many traits are studied in relation to the number of animals used in the reported experiment.

Response: Thank you for your comment. We have addressed the limitations of this study at the end of the “Discussion” section as following: “The limitation of this study is that the sample size used is slightly smaller. Although we have used more stringent standards to screen differentially expressed genes and analyze trait differences, there may still be some bias in the results. Therefore, we will conduct functional experimental verification of some key genes in the future. A sufficient number of samples will be used for experiments in future studies to ensure the credibility and representativeness of the data”. Please refer to the lines 551-556.

The genomic part is adequately described, the statistical analyses are described in a very cursory manner, this can be improved.

Response: Thank you for your suggestion. We have added the information of traits, gene and parameter information used in correlation analysis as following: “Differences in adipose-related traits (CFR, AFR, BF_A, BF_B, BF_C, BF_Avg, BF_67, FWR_F, FWR_M, FWR_H and FWR_T) across comparison combinations were analyzed using a two-tailed t-test” and “These genes include the 182 genes in the tan module of WGCNA, DEGs of Feed A and Feed B groups during the 80 kg body weight period, and DEGs of Feed A and Feed B groups during the 120 kg body weight period. The correlation coefficient and P value were calculated with default parameters.” Please refer to the lines 220-221 and 224-227.

Discussion: In general the discussion is clearly and adequately conducted. I think the authors do an excellent job in describing and discussing their results, but the main concern is in the experimental setup. There are too few animals to make adequate inference from the information that is generated.

Response: Thank you for your comment. Like the answer mentioned above, we have addressed the limitations of this study at the end of the “Discussion” section as following: “The limitation of this study is that the sample size used is slightly smaller. Although we have used more stringent standards to screen differentially expressed genes and analyze trait differences, there may still be some bias in the results. Therefore, we will conduct functional experimental verification of some key genes in the future. A sufficient number of samples will be used for experiments in future studies to ensure the credibility and representativeness of the data”. Please refer to the lines 551-556.

Conclusions should be improved by highlighting the main findings, but not in the form of results. The second part of the conclusions seems to me to be in a better form.

Response: Thank you for your suggestion. We have changed the sentence “Some adipose-deposition-related genes (ACSL1, ANGPTL8, CCNP, CD248, CKB, CYP4A24, ETV4, FABP3, FABP7, ME1, PER2, PLIN5, PPARD, PTGES, RXRG, TRIB3) and pathways (PPAR signaling pathway, Insulin signaling pathway, PI3K-Akt signaling pathway, Wnt signaling pathway, MAPK signaling pathway) may contribute to adipose deposition.” to “This process is regulated by transcriptome and single-cell transcriptome. Although we identified some genes and pathways related to adipose deposition.” in the revised manuscript. Please refer to the lines 560-562.

Thank you very much for the high quality of the peer-review. We hope that our revisions have meticulously addressed all the comments raised here and will be acceptable for publication in Genes.

Yours sincerely,

Xinxing Dong, Ph.D.

College of Animal Science and Technology, Yunnan Agricultural University,

No. 95   Jinhei Road,

Panlong District, 650201 Kunming, P. R. China

Tel: +86-13529185971

Reviewer 2 Report

Comments and Suggestions for Authors

It is a well-structured and statistically well-analyzed work with appropriate tools for transcriptomic data. However, it can improve if these observations are attended to:

It is important to focus the discussion by mentioning that walnut cake not only contains proteins, and not only assigning the expression levels to the proteins of the walnut, it has been reported in other works that the compounds in the residues, such as polyunsaturated fatty acids , phytohormones, sugars and antioxidants, can induce changes in expression in various animal tissues.

Line 66: include in some section of the document, the nutritional table of soy waste vs. walnut waste.

Line 75: why did they use the minimum of 3 biological replicates  (3 animals) per treatment?

Line 97: table 2, it is confusing for the reader, 8-15 kg, 15-30 kg, 30-60 kg and 60-120 kg. Do these values correspond to the sacrifice weight of the animals after finishing the experiment or the weight at the beginning of the experiment?

Line 97: table 2, it is confusing for the reader, for each treatment (8-15 kg) there are two treatments feed A and Fedd B, if it occurs up to 60-120 kg. For each treatment, how many animals did they use? Above you describe that you used 12 animals, and 3 for each treatment, and in this table, you report four treatments with two replicates, there are more than 12 animals, please clarify this.

Line 98: missing to indicate the RIN values of the RNA samples

Line 111: A table is missing, reporting the quality values of the readings for each sample.

Line 108: Were 12 libraries and transcriptomes processed?

Line 113: Was the STAR software to assemble the reads or compare with the reference genome?

Line 117: Did you calculate the false positive rate?

Table 1A. The diagram with the pigs helps to resolve the previous doubt about the treatments. I consider that this part of the figure should be placed in the first part of the methodology.

Line 217: 153 and 225 up-regulated, 153 for food A and 225 for food B?.

Line 426; change the focus from "protein substitute" to nut flour. Because within the nut cake, in addition to proteins, it also includes other compounds such as polyunsaturated fatty acids, which can modify the expression of genes. By itself, the protein induces significant changes in expression but it is also induced by other compounds that accompany the diet such as different types of fatty acids, sugars, antioxidants, etc.

Line 442: include in the paragraph, there is also evidence that chia residues (salvia hispanica) improve the immune system in rabbits (jimenez-rojas et al 2018).

Reviewer 3 Report

Comments and Suggestions for Authors

Peer review report on: “Transcriptomic analyses reveal the effects of walnut kernel cake on adipose deposition in pig”

General comments:

The use of biowastes in animal feeding is a very interesting area and one that is worthy of experimental attention. These by-products, in addition to reducing the environmental impact of the food production chain, can also improve product quality. The authors researched the link between the walnut kernel as a dietary protein substitute and its effects on adipose transcriptome levels using transcriptomics and bioinformatics tools.

The chapters of the manuscript are mostly fair and well structured, the paper is generally clear and well written, and the information obtained can deliver useful sights for the readers.

I have a couple suggestions, comments and questions:

Line 99: Could you briefly add some information about protocol of RNA isolation? Or a reference with the details of protocol?

Line 84: Why did you choose the 80 kg and 120 kg stages to collect adipose tissue?

Line 99-109: Information of manufacturers are missing.

Figure 1 B and C: Are the data presenting as mean±SEM or SD?

Line 222: Please explain, that at the stages of 80 kg and 120 kg there were 44 shared DEGs, of which 14 shared up-regulated DEGs and 12 shared down-regulated. 14 up-regulated and 12 down-regulated DEGs represent 36 DEGs, not 44.

Line 240: What does BP mean?

Figure 3 A, E: What do the different colours mean?

Round 2

Reviewer 1 Report

Comments and Suggestions for Authors

The authors carried out a detailed revision of the paper incorporating the reviewers' recommendations. The conclusions were improved as well as the tables. My suggestion is to accept the document in this revised and improved version.